# AUTOMATED CONCATENATION OF EMBEDDINGS FOR STRUCTURED PREDICTION

## ABSTRACT

Pretrained contextualized embeddings are powerful word representations for structured prediction tasks. Recent work found that better word representations can be obtained by concatenating different types of embeddings. However, the selection of embeddings to form the best concatenated representation usually varies depending on the task and the collection of candidate embeddings, and the ever-increasing number of embedding types makes it a more difficult problem. In this paper, we propose Automated Concatenation of Embeddings (ACE) to automate the process of finding better concatenations of embeddings for structured prediction tasks, based on a formulation inspired by recent progress on neural architecture search. Specifically, a controller alternately samples a concatenation of embeddings, according to its current belief of the effectiveness of individual embedding types in consideration for a task, and updates the belief based on a reward. We follow strategies in reinforcement learning to optimize the parameters of the controller and compute the reward based on the accuracy of a task model, which is fed with the sampled concatenation as input and trained on a task dataset. Empirical results on 6 tasks and 21 datasets show that our approach outperforms strong baselines and achieves state-of-the-art performance with fine-tuned embeddings in the vast majority of evaluations.

## 1 INTRODUCTION

Recent developments on pretrained contextualized embeddings have significantly improved the performance of structured prediction tasks in natural language processing. Approaches based on contextualized embeddings, such as ELMo (Peters et al., 2018), Flair (Akbik et al., 2018), BERT (Devlin et al., 2019), and XLM-R (Conneau et al., 2020), have been consistently raising the state-of-the-art for various structured prediction tasks. Concurrently, research has also showed that word representations based on the concatenation of multiple pretrained contextualized embeddings and traditional non-contextualized embeddings (such as word2vec (Mikolov et al., 2013) and character embeddings (Santos & Zadrozny, 2014)) can further improve performance (Peters et al., 2018; Akbik et al., 2018; Straková et al., 2019; He & Choi, 2020). Given the ever-increasing number of embedding learning methods that operate on different granularities (e.g., word, subword, or character level) and with different model architectures, choosing the best embeddings to concatenate for a specific task becomes non-trivial, and exploring all possible concatenations can be prohibitively demanding in computing resources.

Neural architecture search (NAS) is an active area of research in deep learning to automatically search for better model architectures, and has achieved state-of-the-art performance on various tasks in computer vision, such as image classification (Real et al., 2019), semantic segmentation (Liu et al., 2019a), and object detection (Ghiasi et al., 2019). In natural language processing, NAS has been successfully applied to find better RNN structures (Zoph & Le, 2017; Pham et al., 2018b) and recently better transformer structures (So et al., 2019; Zhu et al., 2020). In this paper, we propose the Automated Concatenation of Embeddings (ACE) approach to automate the process of finding better concatenations of embeddings for structured prediction tasks, formulated as an NAS problem. In this approach, an iterative search process is guided by a controller based on its belief that models the effectiveness of individual embedding candidates in consideration for a specific task. At each step, the controller samples a concatenation of embeddings according to the belief model and feeds the concatenated word representations as inputs to a task model, which in turn is trained on the

task dataset and returns the model accuracy as a reward signal to update the belief model. We use the policy gradient algorithm (Williams, 1992) in reinforcement learning (Sutton & Barto, 1992) to solve the optimization problem. In order to improve the efficiency of the search process, we also design a special reward function by accumulating all the rewards based on the transformation between the current concatenation and all previously sampled concatenations.

Our approach is different from previous work on NAS in the following aspects:

1. Unlike most previous work, we focus on searching for better word representations rather than better model architectures.

2. We design a unique search space for the embedding concatenation search. Instead of using RNN as in previous work of Zoph & Le (2017), we design a more straightforward controller to generate the embedding concatenation. We design a novel reward function in the objective of optimization to better evaluate the effectiveness of each concatenated embeddings.

3. Our approach is efficient and practical. ACE can find a strong word representation on a single GPU with only a few GPU-hours for structured prediction tasks, while a lot of the NAS approaches require dozens of or even thousands of GPU-hours to search for good neural architecture.

4. The task model from ACE achieves high accuracy without the need for retraining, while in previous work of NAS the resulting neural network usually requires retraining from scratch.

Empirical results show that ACE outperforms strong baselines. Furthermore, we show that when ACE is applied to concatenate pretrained contextualized embeddings which are already fine-tuned on specific tasks, we can achieve state-of-the-art or competitive accuracy on 6 structured prediction tasks including Named Entity Recognition (Sundheim, 1995), Part-Of-Speech tagging (DeRose, 1988), chunking (Tjong Kim Sang & Buchholz, 2000), aspect extraction (Hu & Liu, 2004), syntactic dependency parsing (Tesnière, 1959) and semantic dependency parsing (Oepen et al., 2014) over 21 datasets. Besides, we also analyze the advantage of ACE and reward function design over the baselines and show the advantage of ACE over ensemble models.

## 2 RELATED WORK

### 2.1 EMBEDDINGS

Non-contextualized embeddings, such as word2vec (Mikolov et al., 2013), GloVe (Pennington et al., 2014), and fastText (Bojanowski et al., 2017), help lots of NLP tasks. Character embeddings (Santos & Zadrozny, 2014) are trained together with the task and applied in many structured prediction tasks (Ma & Hovy, 2016; Lample et al., 2016; Dozat & Manning, 2018). For pretrained contextualized embeddings, ELMo (Peters et al., 2018), a pretrained contextualized word embedding generated with multiple Bidirectional LSTM layers, significantly outperforms previous state-of-the-art approaches on several NLP tasks. Following this idea, Akbik et al. (2018) proposed Flair embeddings, which is a kind of contextualized character embeddings and achieved strong performance in sequence labeling tasks. Recently, Devlin et al. (2019) proposed BERT, which encodes contextualized sub-word information by Transformers and significantly improves the performance on a lot of NLP tasks. Much research such as RoBERTa (Liu et al., 2019c) has focused on improving BERT model's performance through stronger masking strategies. Moreover, multilingual contextualized embeddings become popular. Pires et al. (2019) and Wu & Dredze (2019) showed that Multilingual BERT (M-BERT) could learn a good multilingual representation effectively with strong cross-lingual zero-shot transfer performance in various tasks. Conneau et al. (2020) proposed XLM-R, which is trained on a larger multilingual corpus and significantly outperforms M-BERT on various multilingual tasks.

### 2.2 NEURAL ARCHITECTURE SEARCH

Recent progress on deep learning has shown that network architecture design is crucial to the model performance. However, designing a strong neural architecture for each task requires enormous efforts, high level of knowledge, and experiences over the task domain. Therefore, automatic design of neural architecture is desired. A crucial part of NAS is search space design, which defines

the discoverable NAS space. Previous work (Baker et al., 2017; Zoph & Le, 2017; Xie & Yuille, 2017) designs a global search space (Elsken et al., 2019) which incorporates structures from hand-crafted architectures. For example, Zoph & Le (2017) designed a chained-structured search space with skip connections. The global search space usually has a considerable degree of freedom. As an example, the approach of Zoph & Le (2017) takes 22,400 GPU-hours to search on CIFAR-10 dataset. Based on the observation that existing hand-crafted architectures contain repeated structures (Szegedy et al., 2016; He et al., 2016; Huang et al., 2017), Zoph et al. (2018) explored cell-based search space which can reduce the search time to 2,000 GPU-hours.

In recent NAS research, reinforcement learning and evolutionary algorithms are the most usual approaches. In reinforcement learning, the agent's actions are the generation of neural architectures and the action space is identical to the search space. Previous work usually applies an RNN layer (Zoph & Le, 2017; Zhong et al., 2018; Zoph et al., 2018) or use Markov Decision Process (Baker et al., 2017) to decide the hyper-parameter of each structure and decide the input order of each structure. Evolutionary algorithms have been applied to architecture search for many decades (Miller et al., 1989; Angeline et al., 1994; Stanley & Miikkulainen, 2002; Floreano et al., 2008; Jozefowicz et al., 2015). The algorithm repeatedly generates new populations through recombination and mutation operations and selects survivors through competing among the population. Recent work with evolutionary algorithms differ in the method on parent/survivor selection and population generation. For example, Real et al. (2017), Liu et al. (2018a), Wistuba (2018) and Real et al. (2019) applied tournament selection (Goldberg & Deb, 1991) for the parent selection while Xie & Yuille (2017) keeps all parents. Suganuma et al. (2017) and Elsken et al. (2018) chose the best model while Real et al. (2019) chose several latest models as survivors.

## 3 AUTOMATED CONCATENATION OF EMBEDDINGS

In ACE, a task model and a controller interact with each other repeatedly. The task model predicts the task output, while the controller searches for better embedding concatenation as the word representation for the task model to achieve higher accuracy. Given an embedding concatenation generated from the controller, the task model is trained over the task data and returns a reward to the controller. The controller receives the reward to update its parameter and samples a new embedding concatenation for the task model. Figure 1 shows the general architecture of our approach.

### 3.1 TASK MODEL

For the tasks model, we emphasis on sequence-structured and graph-structured outputs. Given a structured prediction task with input sentence $x$ and structured output $y$, we can calculate the probability distribution $P(y|x)$ by:

$$P(y|x) = \frac{\exp\left(\text{Score}(x, y)\right)}{\sum_{y' \in \mathbb{Y}(x)} \exp\left(\text{Score}(x, y')\right)}$$

where $\mathbb{Y}(x)$ represents all possible output structures given the input sentence $x$. Depending on different structured prediction tasks, the output structure $y$ can be label sequences, trees, graphs or other structures. In this paper, we use sequence-structured and graph-structured outputs as two exemplar structured prediction tasks. We use BiLSTM-CRF model (Ma & Hovy, 2016; Lample et al., 2016) for sequence-structured outputs and use BiLSTM-Biaffine model (Dozat & Manning, 2017) for graph-structured outputs:

$$P^{\text{seq}}(y|x) = \text{BiLSTM-CRF}(V, y); \quad P^{\text{graph}}(y|x) = \text{BiLSTM-Biaffine}(V, y)$$

where $V = [v_1; \cdots; v_n]$, $V \in \mathbb{R}^{d \times n}$ is a matrix of the word representations for the input sentence $x$ with $n$ words, $d$ is the hidden size of the concatenation of all embeddings. The word representation $v_i$ of $i$-th word is a concatenation of $L$ types of word embeddings:

$$v_i^l = \text{embed}_i^l(x); \quad v_i = [v_i^1; v_i^2; \ldots; v_i^L]$$

where embed$^l$ is the model of $l$-th embeddings, $v_i \in \mathbb{R}^d$, $v_i^l \in \mathbb{R}^{d^l}$. $d^l$ is the hidden size of embed$^l$.

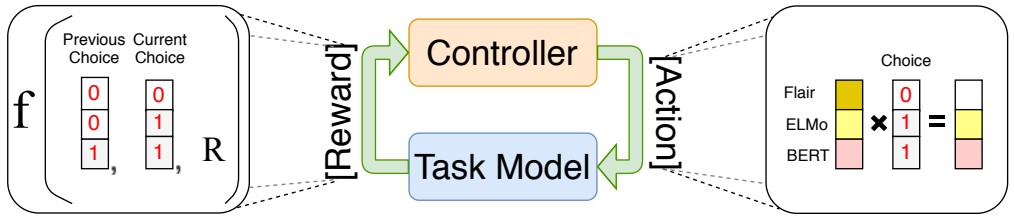

Figure 1: The main paradigm of our approach is shown in the middle, where an example of reward function is represented in the left and an example of a concatenation action is shown in the right.

## 3.2 SEARCH SPACE DESIGN

The neural architecture search space can be represented as a set of neural networks (Elsken et al., 2019). A neural network can be represented as a directed acyclic graph with a set of nodes and directed edges. Each node represents an operation, while each edge represents the inputs and outputs between these nodes. In ACE, we represent each embedding candidate as a node. The input to the nodes is the input sentence $x$, and the outputs are the embeddings $v^l$. Since we concatenate the embeddings as the word representation of the task model, there is no connection between nodes in our search space. Without considering the connections between nodes, the search space can be significantly reduced. For each node, there are a lot of options to extract word features. Taking BERT embeddings as an example, Devlin et al. (2019) concatenated the last four layers as word features while Kondratyuk & Straka (2019) applied a weighted sum of all twelve layers. However, the empirical results (Devlin et al., 2019) do not show a significant difference in accuracy. We follow the typical usage for each embedding to further reduce the search space. As a result, each embedding only has a fixed operation and the resulting search space contains $2^L - 1$ possible combinations of nodes.

In NAS, weight sharing (Pham et al., 2018a) shares the weight of structures in training different neural architectures to reduce the training cost. In comparison, we fixed the weight of pretrained embedding candidates in ACE except for the character embeddings. Instead of sharing the parameters of the embeddings, we share the parameters of the task models at each step of search. However, the hidden size of word representation varies over the concatenations, making the weight sharing of structured prediction models difficult. Instead of deciding whether each node exists in the graph, we keep all nodes in the search space and add an additional operation for each node to indicate whether the embedding is masked out. To represent the selected concatenation, we use a binary vector $a = [a_1, \cdots, a_l, \cdots, a_L]$ as an mask to mask out the embeddings which are not selected:

$$v_i = [v_i^1 a_1; \ldots; v_i^l a_l; \ldots; v_i^L a_L] \tag{1}$$

where $a_l$ is a binary variable. Since the input $V$ is applied to a linear layer in the BiLSTM layer, multiplying the mask with the embeddings is equivalent to directly concatenating the selected embeddings:

$$W = [W_1; W_2; \ldots; W_L]; \quad W^\top v_i = \sum_{l=1}^{L} W_l^\top v_i^l a_l; \quad W \in \mathbb{R}^{d \times h} \text{ and } W_l \in \mathbb{R}^{d^l \times h} \tag{2}$$

Therefore, the model weights can be shared after applying the embedding mask to all embedding candidates' concatenation. Another benefit of our search space design is that we can remove the unused embedding candidates and the corresponding weights in $W$ for a lighter task model after the best concatenation is found by ACE.

## 3.3 SEARCHING IN THE SPACE

During search, the controller generates the embedding mask for the task model iteratively. We use parameters $\theta = [\theta_1; \theta_2; \ldots; \theta_L]$ for the controller instead of the RNN structure applied in previous approaches (Zoph & Le, 2017; Zoph et al., 2018). The probability distribution of selecting an concatenation $a$ is $P^{\text{ctrl}}(a; \theta) = \prod_{l=1}^{L} P_l^{\text{ctrl}}(a_l; \theta_l)$. Each element $a_l$ of $a$ is sampled independently from a Bernoulli distribution, which is defined as:

$$P_l^{\text{ctrl}}(a_l = 1; \theta_l) = \sigma(\theta_l); \quad P_l^{\text{ctrl}}(a_l = 0; \theta_l) = 1 - P_l^{\text{ctrl}}(a_l = 1; \theta_l) \tag{3}$$

where $\sigma$ is the sigmoid function. Given the mask, the task model is trained until convergence and returns an accuracy $R$ on the development set. As the accuracy cannot be back-propagated to the controller, we use the reinforcement algorithm for optimization. The accuracy $R$ is used as the reward signal to train the controller. The controller's target is maximizing the expected reward $J(\boldsymbol{\theta}) = \mathbb{E}_{P^{\mathrm{ctrl}}(\boldsymbol{a};\boldsymbol{\theta})}[R]$ through the policy gradient method (Williams, 1992). In our approach, since calculating the exact expectation is intractable, the gradient of $J(\boldsymbol{\theta})$ is approximated by sampling only one selection following the distribution $P^{\mathrm{ctrl}}(\boldsymbol{a};\boldsymbol{\theta})$ at each step for training efficiency:

$$\nabla_{\boldsymbol{\theta}} J(\boldsymbol{\theta}) \approx \sum_{l=1}^{L} \nabla_{\boldsymbol{\theta}} \log P_l^{\mathrm{ctrl}}(a_l;\theta_l)(R - b) \tag{4}$$

where $b$ is the baseline function to reduce the high variance of the update function. The baseline usually can be the highest accuracy during the search process. Instead of merely using the highest accuracy of development set over the search process as the baseline, we design a reward function on how each embedding candidate contributes to accuracy change by utilizing all searched concatenations' development scores. We use a binary vector $|\boldsymbol{a}^t - \boldsymbol{a}^i|$ to represent the change between current embedding concatenation $\boldsymbol{a}^t$ at current time step $t$ and $\boldsymbol{a}^i$ at previous time step $i$. We then define the reward function as:

$$\boldsymbol{r}^t = \sum_{i=1}^{t-1}(R_t - R_i)|\boldsymbol{a}^t - \boldsymbol{a}^i| \tag{5}$$

where $\boldsymbol{r}^t$ is a vector with length $L$ representing the reward of each embedding candidate. $R_t$ and $R_i$ are the reward at time step $t$ and $i$. When the Hamming distance of two concatenations $Hamm(\boldsymbol{a}^t, \boldsymbol{a}^i)$ gets larger, the changed candidates' contribution to the accuracy becomes less noticeable. The controller may be misled to reward a candidate that is not very helpful. We apply a discount factor to reduce the reward for two concatenations with a large Hamming distance to alleviate this issue. Our final reward function is:

$$\boldsymbol{r}^t = \sum_{i=1}^{t-1}(R_t - R_i)\gamma^{Hamm(\boldsymbol{a}^t, \boldsymbol{a}^i)-1}|\boldsymbol{a}^t - \boldsymbol{a}^i| \tag{6}$$

where $\gamma \in (0, 1)$. Eq. 4 is then reformulated as:

$$\nabla_{\boldsymbol{\theta}} J_t(\boldsymbol{\theta}) \approx \sum_{l=1}^{L} \nabla_{\boldsymbol{\theta}} \log P_l^{\mathrm{ctrl}}(a_l^t;\theta_l)r_l^t \tag{7}$$

### 3.4 TRAINING

To train the controller, we use a dictionary $\mathbb{D}$ to store the concatenations and validation scores. At $t = 1$, we train the task model with all embedding candidates concatenated. From $t = 2$, we repeat the following steps until a maximum iteration $T$:

- Sample a concatenation $\boldsymbol{a}^t$ based on the probability distribution in Eq. 3.
- Train the task model with $\boldsymbol{a}^t$ following Eq. 1 and evaluate the model on the development set to get the accuracy $R_t$.
- Given the concatenation $\boldsymbol{a}^t$, accuracy $R_t$ and $\mathbb{D}$, compute the gradient of the controller following Eq. 7 and update the parameters of controller.
- Add $\boldsymbol{a}^t$ and $R_t$ into $\mathbb{D}$, set $t = t + 1$.

When sampling $\boldsymbol{a}^t$, we avoid selecting the previous concatenation $\boldsymbol{a}^{t-1}$ and the all-zero vector (i.e., selecting no embedding). If $\boldsymbol{a}^t$ is in the dictionary $\mathbb{D}$, we compare the $R_t$ with the value in the dictionary and keep the highest one.

## 4 EXPERIMENTS

### 4.1 DATASETS AND CONFIGURATIONS

To show ACE's effectiveness, we conduct extensive experiments on a variety of structured prediction tasks varying from syntactic tasks to semantic tasks. The tasks are named entity recognition (NER),

Part-Of-Speech (POS) tagging, Chunking, Aspect Extraction (AE), Syntactic Dependency Parsing (DP) and Semantic Dependency Parsing (SDP). The details of the tasks are in Appendix A.1.

We train the controller for 30 steps and save the task model with the highest accuracy on the development set as the final model for testing. For all experiments, we report the averaged accuracy of 3 runs. For other settings, please refer to Appendix A.3 for more details.

## 4.2 Embeddings

**Basic Settings:** For the candidates of embeddings on English datasets, we use the language-specific model for ELMo, Flair, base BERT, GloVe word embeddings, fastText word embeddings, non-contextual character embeddings (Lample et al., 2016), multilingual Flair (M-Flair), M-BERT and XLM-R embeddings. The size of the search space in our experiments is $2^{11} - 1 = 2047^1$. For language-specific models of other languages, please refer to Section A.4 for more details. In AE, there is no available language-specific BERT, Flair and ELMo embeddings for Russian and there is no available language-specific Flair and ELMo embeddings for Turkish. We use the corresponding English embeddings instead so that the search spaces of these datasets are almost identical to those of the other datasets. All embeddings are fixed during training except that the character embeddings are trained over the task. The empirical results are reported in Section 4.3.1.

**Embedding Fine-tuning:** Fine-tuning transformer-based embeddings is a usual approach to get better accuracy. In sequence labeling, most of the work follows the fine-tuning pipeline of BERT that connects the BERT model with a linear layer for word-level classification. However, when multiple embeddings are concatenated, fine-tuning a specific group of embeddings becomes difficult. It is impractical to train multiple embeddings because of complicated hyper-parameter settings and massive GPU memory consumption. To alleviate this problem, we first fine-tune the transformer-based embeddings with the task and then concatenate these embeddings together with other embeddings in the basic setting to apply ACE. The empirical results are reported in Section 4.3.2.

Table 1: Comparison with concatenating all embeddings and random search baselines on 6 tasks.

| | NER | | | | POS | | | AE | | | | | | | |
|---|---|---|---|---|---|---|---|---|---|---|---|---|---|---|---|
| | de | en | es | nl | Ritter | ARK | TB-v2 | 14Lap | 14Res | 15Res | 16Res | es | nl | ru | tr |
| **ALL** | 83.1 | 92.4 | **88.9** | 89.8 | 90.6 | 92.1 | 94.6 | 82.7 | 88.5 | 74.2 | 73.2 | 74.6 | 75.0 | 67.1 | 67.5 |
| **RANDOM** | 84.0 | 92.6 | 88.8 | 91.9 | 91.3 | 92.6 | 94.6 | 83.6 | 88.1 | 73.5 | 74.7 | 75.0 | 73.6 | 68.0 | 70.0 |
| **ACE** | **84.2** | **93.0** | **88.9** | **92.1** | **91.7** | **92.8** | **94.8** | **83.9** | **88.6** | **74.9** | **75.6** | **75.7** | **75.3** | **70.6** | **71.1** |

| | CHUNK | DP | | SDP | | | | | | | AVG |
|---|---|---|---|---|---|---|---|---|---|---|---|
| | CoNLL 2000 | UAS | LAS | DM-ID | DM-OOD | PAS-ID | PAS-OOD | PSD-ID | PSD-OOD | | |
| **ALL** | 96.7 | 96.7 | 95.1 | 94.3 | 90.8 | **94.6** | 92.9 | 82.4 | 81.7 | | 85.3 |
| **RANDOM** | 96.7 | 96.8 | 95.2 | 94.4 | 90.8 | **94.6** | 93.0 | 82.3 | 81.8 | | 85.7 |
| **ACE** | **96.8** | **96.9** | **95.3** | **94.5** | **90.9** | 94.5 | **93.1** | **82.5** | **82.1** | | **86.2** |

## 4.3 Results

We use the following abbreviations in our experiments: **UAS**: Unlabeled Attachment Score; **LAS**: Labeled Attachment Score; **ID**: In-domain test set; **OOD**: Out-of-domain test set; **F & G (2019)**: Fernández-González & Gómez-Rodríguez (2019); **F & G (2020)**: Fernández-González & Gómez-Rodríguez (2020); **D & M (2018)**: Dozat & Manning (2018). In all tables, we use ISO 639-1 language codes to represent each language.

### 4.3.1 Comparison With Baselines

To show the effectiveness of our approach, we compare our approach with two strong baselines. For the first one, we let the task model learn by itself the contribution of each embedding candidate that is helpful to the task. We set $a$ to all-ones (i.e., the concatenation of all the embeddings) and train the task model (All). The linear layer weight $W$ in Eq. 2 reflects the contribution of each candidate. For the second one, we use the random search (Random), a strong baseline in NAS (Li &

---

[1]Flair embeddings have two models (forward and backward) for each language.

Table 2: Comparison with state-of-the-art approaches in NER and POS tagging. [†]: Models are trained on both train and development set. [‡]: Models are trained with document information. [◇]: Results are from Conneau et al. (2020).

| | | | NER | | | | | POS | | |
|---|---|---|---|---|---|---|---|---|---|---|
| | de | $de_{06}$ | en | es | nl | | | Ritter | ARK | TB-v2 |
| Akbik et al. (2018)[†] | - | 88.3 | 93.1 | - | - | Owoputi et al. (2013) | | 90.4 | 93.2 | 94.6 |
| Baevski et al. (2019) | - | - | **93.5** | - | - | Gui et al. (2017) | | 90.9 | - | 92.8 |
| Straková et al. (2019)[†] | 85.1 | - | 93.4 | 88.8 | 92.7 | Gui et al. (2018) | | 91.2 | 92.4 | - |
| Yu et al. (2020)[†‡] | 86.4 | 90.3 | **93.5** | 90.3 | 93.7 | Nguyen et al. (2020) | | 90.1 | **94.1** | 95.2 |
| XLM-R+Fine-tune[◇] | 85.8 | - | 92.9 | 89.7 | 92.5 | XLM-R+Fine-tune | | 93.0 | 93.4 | 95.0 |
| ACE+Fine-tune | **87.0** | **90.5** | **93.5** | **91.7** | **94.6** | ACE+Fine-tune | | **93.4** | 93.8 | **95.6** |

Table 3: Comparison with state-of-the-art approaches in chunking and aspect extraction. [†]: We report the results reproduced by Wei et al. (2020).

| | **CHUNK** | | | **AE** | | | | | | | |
|---|---|---|---|---|---|---|---|---|---|---|---|
| | CoNLL 2000 | | | 14Lap | 14Res | 15Res | 16Res | es | nl | ru | tr |
| Akbik et al. (2018) | 96.7 | | Xu et al. (2018)[†] | 84.2 | 84.6 | 72.0 | 75.4 | - | - | - | - |
| Clark et al. (2018) | 97.0 | | Xu et al. (2019) | 84.3 | - | - | 78.0 | - | - | - | - |
| Liu et al. (2019b) | **97.3** | | Wang et al. (2020a) | - | - | - | 72.8 | 74.3 | 72.9 | 71.8 | 59.3 |
| Wang et al. (2020b) | - | | Wei et al. (2020) | 82.7 | 87.1 | 72.7 | 77.7 | - | - | - | - |
| XLM-R+Fine-tune | 96.5 | | XLM-R+Fine-tune | 81.3 | 88.4 | 77.3 | 78.5 | 77.8 | 72.1 | 75.7 | 66.7 |
| ACE+Fine-tune | 97.0 | | ACE+Fine-tune | **85.0** | **89.8** | **78.5** | **81.2** | **78.8** | **76.7** | **76.7** | **77.7** |

Talwalkar, 2020). For `Random`, we run the same maximum iteration as in ACE. Table 1 shows that ACE outperforms both baselines in 6 tasks over 23 test sets with only two exceptions. Comparing `Random` with `All`, `Random` outperforms `All` by 0.4 on average and surpasses the accuracy of `All` on 14 out of 23 test sets, which shows that concatenating all embeddings may not be the best solution to most structured prediction tasks. In general, searching for the concatenation for the word representation is essential in most cases, and our search design can usually lead to better results compared to both of the baselines.

### 4.3.2 COMPARISON WITH STATE-OF-THE-ART APPROACHES

As we have shown, ACE has an advantage in searching for better embedding concatenations. We further show that ACE is competitive or even stronger than state-of-the-art approaches. In some tasks, We have several additional settings to better compare with previous work. In NER, we also conduct a comparison on the revised version of German datasets in the CoNLL 2006 shared task (Buchholz & Marsi, 2006). In parsing tasks, we use XLNet (Yang et al., 2019), which is significantly stronger than BERT in DP (Zhou & Zhao, 2019). In SDP, the state-of-the-art approaches used POS tags and lemmas as additional word features to the network. We add these two features to the embedding candidates and train the embeddings together with the task. We use the fine-tuned BERT embeddings, XLM-R embeddings, and XLNet embeddings on each task instead of the pretrained version of these embeddings as the candidates. For the NER tasks, we use the XLM-R models fine-tuned by Hugging Face instead.[2]

We additionally compare with fine-tuned XLM-R model for NER, POS tagging, chunking and AE, and compare with fine-tuned XLNet model for DP and SDP, which are strong fine-tuned models in most of the experiments. Results are shown in Table 2, 3, 4. Results show that ACE with fine-tuned embeddings achieves state-of-the-art performance in 22 out of 24 test sets and is competitive with the state-of-the-art approaches in the other 2 test sets. Our approach is competitive or even stronger than the approaches using additional information in NER and DP, which shows that finding a good word representation helps structured prediction tasks. We also show that ACE is stronger than the fine-tuned models, which shows the effectiveness of concatenating the fine-tuned embeddings.

---

[2]Please refer to Appendix A.4 and B.1 for more details.

Table 4: Comparison with state-of-the-art approaches in DP and SDP. $^\dagger$: For reference, they additionally used constituency dependencies in training. $^\ddagger$: For reference, we confirmed with the authors of He & Choi (2020) that they used a different data pre-processing script with previous work.

| | DP | | | SDP | | | | | |
| | PTB | | | DM | | PAS | | PSD | |
| | UAS | LAS | | ID | OOD | ID | OOD | ID | OOD |
|---|---|---|---|---|---|---|---|---|---|
| Zhou & Zhao (2019)$^\dagger$ | 97.2 | 95.7 | He & Choi (2020)$^\ddagger$ | 94.6 | 90.8 | 96.1 | 94.4 | 86.8 | 79.5 |
| F & G (2019) | 96.0 | 94.4 | D & M (2018) | 94.0 | 89.7 | 94.1 | 91.3 | 81.4 | 79.6 |
| He & Choi (2020) | 96.8 | 95.3 | Wang et al. (2019) | 93.7 | 88.9 | 93.9 | 90.6 | 81.0 | 79.4 |
| Zhang et al. (2020) | 96.1 | 94.5 | F & G (2020) | 94.4 | 91.0 | 95.1 | 93.4 | 82.6 | 82.0 |
| XLNet+Fine-tune | 97.0 | 95.4 | XLNet+Fine-tune | 94.9 | 92.0 | 94.8 | 93.4 | 82.6 | 82.2 |
| ACE+Fine-tune | **97.2** | **95.7** | ACE+Fine-tune | **95.3** | **92.6** | **95.3** | **93.9** | **83.6** | **83.2** |

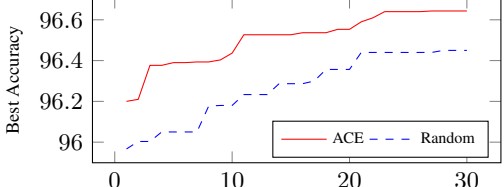 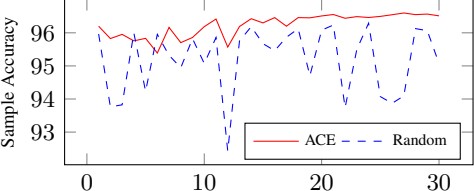

Figure 2: Comparing the efficiency of random search (Random) and ACE. The x-axis is the number of time steps. The left y-axis is the averaged best validation accuracy on CoNLL English NER dataset. The right y-axis is the averaged validation accuracy of the current selection.

## 5 ANALYSIS

### 5.1 EFFICIENCY OF SEARCH METHODS

To show how efficient our approach is compared with the random search algorithm, we compare the algorithm in two aspects on CoNLL English NER dataset. The first aspect is the best development accuracy during training. The left part of Figure 2 shows that ACE is consistently stronger than the random search algorithm in this task. The second aspect is the searched concatenation at each time step. The right part of Figure 2 shows that the accuracy of ACE gradually increases and gets stable when more concatenations are sampled.

### 5.2 ABLATION STUDY ON REWARD FUNCTION DESIGN

To show the effectiveness of the designed reward function, we compare our reward function (Eq. 6) with the reward function without discount factor (Eq. 5) and the traditional reward function (reward term in Eq. 4). We sample 2000 training sentences on CoNLL English NER dataset for faster training and train the controller for 50 steps. Table 5 shows that both the discount factor and the binary vector $|\boldsymbol{a}^t - \boldsymbol{a}^i|$ for the task are helpful in both development and test datasets.

Table 5: Comparison of reward functions.

| | DEV | TEST |
|---|---|---|
| ACE | **93.18** | **90.00** |
| No discount (Eq. 5) | 92.98 | 89.90 |
| Simple (Eq. 4) | 92.89 | 89.82 |

### 5.3 COMPARISON WITH ADDITIONAL BASELINES

We compare ACE with two more approaches to further show the effectiveness of ACE. One is a variant of All, which uses a weighting parameter $\boldsymbol{b} = [b_1, \cdots, b_l, \cdots, b_L]$ passing through a sigmoid function to weight each embedding candidate. Such an approach can explicitly learn the weight of each embedding in training instead of a binary mask. We call this approach All+Weight. Another one is model ensemble, which trains the task model with each embedding candidate individually and uses the trained models to make joint prediction on the test set. We use voting for ensemble as it is simple and fast. For sequence labeling tasks, the models vote for the predicted label at each position. For DP, the models vote for the tree of each sentence. For SDP, the models vote for each potential

Table 6: A comparison among `All`, `Random`, `ACE`, `All+Weight` and `Ensemble`. CHK: chunking.

|  | NER | POS | AE | CHK | DP | | SDP | |
|---|---|---|---|---|---|---|---|---|
|  |  |  |  |  | UAS | LAS | ID | OOD |
| All | 92.4 | 90.6 | 73.2 | 96.7 | 96.7 | 95.1 | 94.3 | 90.8 |
| Random | 92.6 | 91.3 | 74.7 | 96.7 | 96.8 | 95.2 | 94.4 | 90.8 |
| ACE | **93.0** | **91.7** | **75.6** | **96.8** | **96.9** | **95.3** | **94.5** | **90.9** |
| All+Weight | 92.7 | 90.4 | 73.7 | 96.7 | 96.7 | 95.1 | 94.3 | 90.7 |
| Ensemble | 92.2 | 90.6 | 68.1 | 96.5 | 96.1 | 94.3 | 94.1 | 90.3 |
| Ensemble$_{dev}$ | 92.2 | 90.8 | 70.2 | 96.7 | 96.8 | 95.2 | 94.3 | 90.7 |
| Ensemble$_{test}$ | 92.7 | 91.4 | 73.9 | 96.7 | 96.8 | 95.2 | 94.4 | 90.8 |

Table 7: Results of models with document context on NER. +sent/+doc: models with sentence-/document-level embeddings.

|  | de | de$_{06}$ | en | es | nl |
|---|---|---|---|---|---|
| All+sent | 86.8 | 90.1 | 93.3 | 90.0 | 94.4 |
| ACE+sent | 87.1 | 90.5 | 93.6 | 92.4 | 94.6 |
| BERT (2019) | - | - | 92.8 | - | - |
| Akbik et al. (2019) | - | 88.3 | 93.2 | - | 90.4 |
| Yu et al. (2020) | 86.4 | 90.3 | 93.5 | 90.3 | 93.7 |
| All+doc | 87.6 | 91.0 | 93.5 | 93.3 | 93.7 |
| ACE+doc | **88.0** | **91.4** | **94.1** | **95.6** | **95.5** |

labeled arc. We use the confidence of model predictions to break ties if there are more than one agreement with the same counts. We call this approach `Ensemble`. One of the benefits of voting is that it combines the predictions of the task models efficiently without any training process. We can search all possible $2^L - 1$ model ensembles in a short period of time through caching the outputs of the models. Therefore, we search for the best ensemble of models on the development set and then evaluate the best ensemble on the test set (`Ensemble`$_{dev}$). Moreover, we additionally search for the best ensemble on the test set for reference (`Ensemble`$_{test}$), which is the upper bound of the approach. We use the same setting as in Section 4.3.1 and select one of the datasets from each task. For NER, POS tagging, AE, and SDP, we use CoNLL 2003 English, Ritter, 16Res, and DM datasets, respectively. The results are shown in Table 6. Empirical results show that ACE outperforms all the settings of these approaches and even `Ensemble`$_{test}$, which shows the effectiveness of ACE and the limitation of ensemble models. `All`, `All+Weight` and `Ensemble`$_{dev}$ are competitive in most of the cases and there is no clear winner of these approaches on all the datasets. These results show the strength of embedding concatenation. Concatenating the embeddings incorporates information from all the embeddings and forms stronger word representations for the task model, while in model ensemble, it is difficult for the individual task models to affect each other through ensemble.

## 5.4 ACE with Document-Level Representations

Recently, models with document-level word representations extracted from transformer-based embeddings significantly outperform models with sentence-level word representations in NER (Devlin et al., 2019; Yu et al., 2020). To show the effectiveness of ACE with document-level representations, we replace the sentence-level word representations from transformer-based embeddings (i.e., XLM-R and BERT embeddings) with the document-level word representations. We generate document-level representations following Yu et al. (2020). Results are shown in Table 7. We report the test results of `All` to show how the gap between ACE and `All` changes with different kinds of representations. We report the test accuracy of the models with the highest development accuracy following Yu et al. (2020) for a fair comparison. Empirical results show that the document-level representations can significantly improve the accuracy of ACE. Comparing with models with sentence-level representations, the averaged accuracy gap between ACE and `All` is enhanced from 0.7 to 1.1 with document-level representations, which shows that the advantage of ACE becomes stronger with document-level representations.

## 6 Conclusion

In this paper, we propose the Automated Concatenation of Embeddings, which automatically searches for better embedding concatenation for structured prediction tasks. We design a simple search space and use the reinforcement learning with a novel reward function to efficiently guide the controller to search for better embedding concatenations. We take the change of embedding concatenations into the reward function design and show that our new reward function is stronger than the simpler ones. Results show that ACE outperforms strong baselines. Together with fine-tuned embeddings, ACE achieves state-of-the-art performance in 6 tasks over 19 out of 21 datasets.

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

## A    DETAILED CONFIGURATIONS

We use ISO 639-1 language codes to represent languages in the table[3].

---

[3] https://en.wikipedia.org/wiki/List_of_ISO_639-1_codes

### A.1 DATASETS

The details of the 6 structured prediction tasks in our experiments are shown in below:

- **NER**: We use the corpora of 4 languages from the CoNLL 2002 and 2003 shared task (Tjong Kim Sang, 2002; Tjong Kim Sang & De Meulder, 2003) with standard split.

- **POS Tagging**: We use three datasets, Ritter11-T-POS (Ritter et al., 2011), ARK-Twitter (Gimpel et al., 2011; Owoputi et al., 2013) and Tweebank-v2 (Liu et al., 2018b) datasets (Ritter, ARK and TB-v2 in simplification). We follow the dataset split of (Nguyen et al., 2020).

- **Chunking**: We use CoNLL 2000 (Tjong Kim Sang & Buchholz, 2000) for chunking. Since there is no standard development set for CoNLL 2000 dataset, we split 10% of the training data as the development set.

- **Aspect Extraction**: Aspect extraction is a subtask of aspect-based sentiment analysis (Pontiki et al., 2014; 2015; 2016). The datasets are from the laptop and restaurant domain of SemEval 14, restaurant domain of SemEval 15 and restaurant domain of SemEval 16 shared task (14Lap, 14Res, 15Res and 16Res in short). Additionally, we use another 4 languages in the restaurant domain of SemEval 16 to test our approach in multiple languages. We randomly split 10% of the training data as the development set following Li et al. (2019).

- **Syntactic Dependency Parsing**: We use Penn Tree Bank (PTB) 3.0 with the same dataset pre-processing as (Ma et al., 2018).

- **Semantic Dependency Parsing**: We use DM, PAS and PSD datasets for semantic dependency parsing (Oepen et al., 2014) for the SemEval 2015 shared task (Oepen et al., 2015). The three datasets have the same sentences but with different formalisms. We use the standard split for SDP. In the split, there are in-domain test sets and out-of-domain test sets for each dataset.

Among these tasks, NER, POS tagging, chunking and aspect extraction are sequence-structured outputs while dependency parsing and semantic dependency parsing are the graph-sectured outputs. POS Tagging, chunking and DP are syntactic structured prediction tasks while NER, AE, SDP are semantic structured prediction tasks.

### A.2 EVALUATION

To evaluate our models, We use F1 score to evaluate NER, Chunking and AE, use accuracy to evaluate POS Tagging, use unlabeled attachment score (UAS) and labeled attachment score (LAS) to evaluate DP, and use labeled F1 score to evaluate SDP.

### A.3 TASK MODELS AND CONTROLLER

For sequence-structured tasks (i.e., NER, POS tagging, chunking, aspect extraction), we use a batch size of 32 sentences and an SGD optimizer with a learning rate of $0.1$. We anneal the learning rate by $0.5$ when there is no accuracy improvement on the development set for 5 epochs. We set the maximum training epoch to $150$. For graph-structured tasks (i.e., DP and SDP), we use Adam (Kingma & Ba, 2015) to optimize the model with a learning rate of $0.002$. We anneal the learning rate by $0.75$ for every 5000 iterations following Dozat & Manning (2017). We set the maximum training epoch to $300$. For DP, we run the maximum spanning tree McDonald et al. (2005) algorithm to output valid trees in testing. We fix the hyper-parameters of the task models.

We tune the learning rate for the controller among $\{0.1, 0.2, 0.3, 0.4, 0.5\}$ and the discount factor among $\{0.1, 0.3, 0.5, 0.7, 0.9\}$ on the same dataset in Section 5.2. We search for the hyper-parameter through grid search and find a learning rate of $0.1$ and a discount factor of $0.5$ performs the best on the development set. The controller's parameters are initialized to all $0$ so that each candidate is selected evenly in the first two time steps. We use Stochastic Gradient Descent (SGD) to optimize the controller. The training time depends on the task and dataset size. Take the English NER CoNLL dataset as an example. It takes $45$ GPU hours to train the controller for 30 steps on a single Tesla P100 GPU, which is an acceptable training time in practice.

## A.4 SOURCES OF EMBEDDINGS

The sources of the embeddings that we used are listed in Table 8.

Table 8: The embeddings we used in our experiments. The URL is where we downloaded the embeddings. Note that we have confirmed that the XLM-R models fine-tuned on CoNLL 2002/2003 datasets are only trained on the training data.

| EMBEDDING | RESOURCE | URL |
|---|---|---|
| GloVe | Pennington et al. (2014) | `nlp.stanford.edu/projects/glove` |
| fastText | Bojanowski et al. (2017) | `github.com/facebookresearch/fastText` |
| ELMo | Peters et al. (2018) | `github.com/allenai/allennlp` |
| ELMo (Other languages) | Schuster et al. (2019) | `github.com/TalSchuster/CrossLingualContextualEmb` |
| BERT | Devlin et al. (2019) | `huggingface.co/bert-base-cased` |
| M-BERT | Devlin et al. (2019) | `huggingface.co/bert-base-multilingual-cased` |
| BERT (Dutch) | wietsedv | `huggingface.co/wietsedv/bert-base-dutch-cased` |
| BERT (German) | dbmdz | `huggingface.co/bert-base-german-dbmdz-cased` |
| BERT (Spanish) | dccuchile | `huggingface.co/dccuchile/bert-base-spanish-wwm-cased` |
| BERT (Turkish) | dbmdz | `huggingface.co/dbmdz/bert-base-turkish-cased` |
| XLM-R | Conneau et al. (2020) | `huggingface.co/xlm-roberta-large` |
| XLM-R (CoNLL 02 Dutch) | Hugging Face | `huggingface.co/xlm-roberta-large-finetuned-conll02-dutch` |
| XLM-R (CoNLL 02 Spanish) | Hugging Face | `huggingface.co/xlm-roberta-large-finetuned-conll02-spanish` |
| XLM-R (CoNLL 03 English) | Hugging Face | `huggingface.co/xlm-roberta-large-finetuned-conll03-english` |
| XLM-R (CoNLL 03 German) | Hugging Face | `huggingface.co/xlm-roberta-large-finetuned-conll03-german` |
| XLNet | Yang et al. (2019) | `huggingface.co/xlnet-large-cased` |

# B ADDITIONAL ANALYSIS

## B.1 FINE-TUNED MODELS VERSUS ACE

To fine-tune the embeddings, we use AdamW (Loshchilov & Hutter, 2018) optimizer with a learning rate of 5e-5 and trained the contextualized embeddings with the task for 10 epochs. We use a batch size of 32 for BERT, M-BERT and use a batch size of 16 for XLM-R and XLNet. A comparison between ACE and the fine-tuned embeddings that we used in ACE is shown in Table 9, 10. Results show that ACE can further improve the accuracy of fine-tuned models.

Table 9: A comparison between ACE and the fine-tuned embeddings that are used in ACE for NER and POS tagging.

| | NER | | | | | POS | | |
|---|---|---|---|---|---|---|---|---|
| | de | de (Revised) | en | es | nl | Ritter | ARK | TB-v2 |
| BERT+Fine-tune | 78.1 | 82.2 | 91.0 | 83.1 | 83.3 | 91.2 | 91.7 | 94.4 |
| MBERT+Fine-tune | 81.9 | 86.2 | 91.3 | 87.6 | 90.7 | 90.8 | 91.5 | 93.9 |
| XLM-R+Fine-tune | 85.8 | - | 92.9 | 89.7 | 92.5 | 93.0 | 93.4 | 95.0 |
| ACE+Fine-tune | **87.0** | **90.5** | **93.5** | **91.7** | **94.6** | **93.4** | **93.8** | **95.6** |

Table 10: A comparison between ACE and the fine-tuned embeddings we used in ACE for chunking and AE.

| | Chunk | AE | | | | | | | |
|---|---|---|---|---|---|---|---|---|---|
| | CoNLL 2000 | 14Lap | 14Res | 15Res | 16Res | es | nl | ru | tr |
| BERT+Fine-tune | 96.7 | 81.2 | 87.7 | 71.8 | 73.9 | 76.9 | 73.1 | 64.3 | 75.6 |
| MBERT+Fine-tune | 96.6 | 83.5 | 85.0 | 69.5 | 73.6 | 74.5 | 72.6 | 71.6 | 58.8 |
| XLM-R+Fine-tune | 96.5 | 81.3 | 88.4 | 77.3 | 78.5 | 77.8 | 72.1 | 75.7 | 66.7 |
| ACE+Fine-tune | **97.0** | **85.0** | **89.8** | **78.5** | **81.2** | **78.8** | **76.7** | **76.7** | **77.7** |

## B.2 RETRAINING

Most of the work (Zoph & Le, 2017; Zoph et al., 2018; Pham et al., 2018b; So et al., 2019; Zhu et al., 2020) in NAS retrains the searched neural architecture from scratch so that the hyper-parameters of the searched model can be modified or trained on larger datasets. To show whether our searched embedding concatenation is helpful to the task, we retrain the task model with the embedding concatenations on the same dataset from scratch. For the experiment, we use the same dataset settings

Table 11: A comparison between ACE and the fine-tuned embeddings that are used in ACE for DP and SDP.

| | DP | | SDP | | | | | |
| | PTB | | DM | | PAS | | PSD | |
| | UAS | LAS | ID | OOD | ID | OOD | ID | OOD |
|---|---|---|---|---|---|---|---|---|
| BERT+Fine-tune | 96.6 | 95.1 | 94.4 | 91.4 | 94.4 | 93.0 | 82.0 | 81.3 |
| MBERT+Fine-tune | 96.5 | 94.9 | 93.9 | 90.4 | 93.9 | 92.1 | 81.2 | 80.0 |
| XLM-R+Fine-tune | 96.6 | 95.1 | 94.3 | 91.1 | 94.5 | 92.8 | 82.0 | 81.6 |
| XLNET+Fine-tune | 97.0 | 95.4 | 94.9 | 92.0 | 94.8 | 93.4 | 82.6 | 82.2 |
| ACE+Fine-tune | **97.2** | **95.7** | **95.3** | **92.6** | **95.3** | **93.9** | **83.6** | **83.2** |

as in Section 5.3. We train the searched embedding concatenation of each run from ACE 3 times (therefore, 9 runs for each dataset).

Table 12 shows the comparison between retrained models with the searched embedding concatenation from ACE and All. The results show that the retrained models are competitive with ACE in SDP and in chunking. However, in another three tasks, the retrained models perform inferior to ACE, which shows our approach's advantage. The retrained models outperform All in all tasks, which shows the effectiveness of the searched embedding concatenations.

Table 12: A comparison among retrained models, All and ACE. We use the one dataset for each task.

| | NER | POS | Chunk | AE | DP-UAS | DP-LAS | SDP-ID | SDP-OOD |
|---|---|---|---|---|---|---|---|---|
| All | 92.4 | 90.6 | 96.7 | 73.2 | 96.7 | 95.1 | 94.3 | 90.8 |
| Retrain | 92.6 | 90.8 | **96.8** | 73.6 | 96.8 | 95.2 | **94.5** | **90.9** |
| ACE | **93.0** | **91.7** | **96.8** | **75.6** | **96.9** | **95.3** | **94.5** | **90.9** |

### B.3   EFFECT OF EMBEDDINGS IN THE SEARCHED EMBEDDING CONCATENATIONS

There is no clear conclusion on what concatenation of embeddings is helpful to most of the tasks. We analyze the best searched embedding concatenations by ACE over different structured outputs, semantic/syntactic type, and monolingual/multilingual tasks. The percentage of each embedding selected by the best concatenations from all experiments of ACE are shown in Table 13. The best embedding concatenation varies over the output structure, syntactic/semantic level of understanding, and the language. The experimental results show that it is essential to select embeddings for each kind of task separately. However, we also find that the embeddings are strong in specific settings. In comparison to the sequence-structured and graph-structured tasks, we find that M-BERT and ELMo are only frequently selected in sequence-structured tasks while XLM-R embeddings are always selected in graph-structured tasks. For Flair embeddings, the forward and backward model are evenly selected. We suspect one direction of Flair embeddings is strong enough. Therefore concatenating the embeddings from two directions together cannot further improve the accuracy. For non-contextualized embeddings, pretrained word embeddings are frequently selected in sequence-structured tasks, and character embeddings are not. When we dig deeper into the semantic and syntactic type of these two structured outputs, we find that in all best concatenations, BERT embeddings are selected in all syntactic sequence-structured tasks, and Flair, M-Flair, word, and XLM-R embeddings are selected in syntactic graph-structured tasks. In multilingual tasks, all best concatenations in multilingual NER tasks select M-BERT embeddings while M-BERT is rarely selected in multilingual AE tasks. The monolingual Flair embeddings are always selected in NER tasks, and XLM-R is more frequently selected in multilingual tasks than monolingual sequence-structured tasks (**SS**).

Table 13: The percentage of each embedding candidate selected in the best concatenations from ACE. **F** and **MF** are monolingual and multilingual Flair embeddings. We count these two embeddings are selected if one of the forward/backward (**fw/bw**) direction of Flair is selected in the concatenation. We count the **Word** embedding is selected if one of the fastText/GloVe embeddings is selected. **SS**: sequence-structured tasks. **GS**: graph-structured tasks. **Sem.**: Semantic-level tasks. **Syn.**: Syntactic-level tasks. **M-NER**: Multilingual NER tasks. **M-AE**: Multilingual AE tasks. We only use English datasets in **SS** and **GS**. English datasets are removed for **M-NER** and **M-AE**.

| | BERT | M-BERT | Char | ELMo | F | F-bw | F-fw | MF | MF-bw | MF-fw | Word | XLM-R |
|---|---|---|---|---|---|---|---|---|---|---|---|---|
| SS | 0.81 | 0.74 | 0.37 | 0.85 | 0.70 | 0.48 | 0.59 | 0.78 | 0.59 | 0.41 | 0.81 | 0.70 |
| GS | 0.75 | 0.17 | 0.50 | 0.25 | 0.83 | 0.75 | 0.42 | 0.83 | 0.58 | 0.58 | 0.50 | 1.00 |
| Sem. SS | 0.67 | 0.73 | 0.40 | 0.80 | 0.60 | 0.40 | 0.53 | 0.87 | 0.60 | 0.53 | 0.80 | 0.60 |
| Syn. SS | 1.00 | 0.75 | 0.33 | 0.92 | 0.83 | 0.58 | 0.67 | 0.67 | 0.58 | 0.25 | 0.83 | 0.83 |
| Sem. GS | 0.78 | 0.22 | 0.67 | 0.33 | 0.78 | 0.67 | 0.56 | 0.78 | 0.56 | 0.67 | 0.33 | 1.00 |
| Syn. GS | 0.67 | 0.00 | 0.00 | 0.00 | 1.00 | 1.00 | 0.00 | 1.00 | 0.67 | 0.33 | 1.00 | 1.00 |
| M-NER | 0.67 | 1.00 | 0.56 | 0.83 | 1.00 | 0.78 | 1.00 | 0.89 | 0.78 | 0.44 | 0.78 | 0.89 |
| M-AE | 1.00 | 0.33 | 0.75 | 0.33 | 0.58 | 0.42 | 0.42 | 0.75 | 0.25 | 0.75 | 0.50 | 0.92 |

