# OpenReview forum: "Automated Concatenation of Embeddings for Structured Prediction"
_ICLR.cc/2021/Conference — Reject_

### Official Review · AnonReviewer1 · 2020-10-19
**Interesting, good results, but too heavy and complex**

**Rating:** 5
**Confidence:** 3

**Review:**

This paper introduced an interesting application of reinforcement learning in the selection of concatenation of contextual/non-contextual word embeddings.  It is clever to limit the search space on the selection of embedding sources rather than search the whole network structure, as the current strategy is much easier in the training step. The author(s) conducted many experiments that compared many other models (including SOTA models, and ablation study). Those results are pretty good and impressive.

The main concern is the necessity of using the concatenation of those contextual embeddings. Calculating different contextual embeddings will cost many computing resources and affect the model's speed. Instead of using reinforcement learning to learn the concatenation of different embeddings, why not use the ensemble model to aggregate the results from different contextual embedding-based models? For example, we can use three embeddings BERT, ELMO, Glove to build three separate models and then aggerate their predicted results, the computing speed/resource may be similar to the learned embedding concatenation model, but is it possible that the ensemble model outperforms the ACE model? More experiments of the comparison with ensemble models should be conducted to prove the necessity of ACE.

---

> ### Author Response · Authors · 2020-11-22
> **Response to Reviewer #1**
>
> We want to thank you for your feedback and criticisms. We really appreciate you taking the time out to share your thoughtful comments.
>
> ---
> ### For the review:
> We take your suggestion of comparing ACE with ensemble model. We compared our approach with the ensemble model through voting. Furthermore, we can search all possible $2^L-1$ model ensembles in a short period of time through caching the outputs of the models because voting combines the predictions of the task models efficiently without any training process. We searched for all $2^L-1$ possible combinations on development sets and then evaluate the best ensemble on the test set. We additionally search for the best ensemble on the test set for reference, which is the upper bound of the approach. Empirical results show that ACE is stronger than voting in all the cases including searching over all possibilities of ensemble model on the test set. In conclusion, using the concatenation of those contextual embeddings is necessary. Concatenating the embeddings incorporates information from all the embeddings and forms stronger word representations for the task model, while in model ensemble, it is difficult for the individual task models to affect each other through ensemble. Please refer to the first part of our general response to all reviewers and Section 5.3 in our revision for more details of the settings.

---

### Official Review · AnonReviewer2 · 2020-10-28
**Some nice results, but I'm not fully convinced by the technique**

**Rating:** 4
**Confidence:** 4

**Review:**

This paper explores a way of learning how to automatically construct a concatenated set of embeddings for structured prediction tasks in NLP. The paper's model takes up to L embeddings concatenated together and feeds them into standard models (BiLSTM-CRFs or the BiLSTM-Biaffine technique of Dozat and Manning) to tackle problems like POS tagging, NER, dependency parsing, and more.  Search over embedding concaneations is expressed as a search over binary masks of length L.  The controller for this search is parameterized by an independent Bernoulli for each mask position.  The paper's approach learns the controller parameters with policy gradient, where the reward function is (a modified version of) the accuracy on the development set for the given task. This modified reward uses all samples throughout training to effectively get a more fine-grained baseline for the current timestep based on prior samples.  Notably, the paper uses embeddings that are already fine-tuned for each task, as fine-tuning the concatenated embeddings is hard due to divergent step sizes and steep computational requirements.

Results show gains over randomly searching the space of binary masks. The overall model outperforms XLM-R in a range of multilingual settings.

This paper has some nice empirical results and the simplicity of its approach is attractive. But there are two shortcomings of the paper I will discuss.

MOTIVATION/COMPARISONS

The authors motivate their technique by drawing parallels to neural architecture search. But I actually think what the authors are doing more closely resembles ensembling, system combination, or model stacking, e.g.:
https://www.aclweb.org/anthology/N09-2064.pdf
https://www.aclweb.org/anthology/N18-1201.pdf

When you take large Transformer models (I have to imagine that the Transformers are contributing more to the performance than GloVe and other static word embeddings -- and Table 11 supports this somewhat) and staple a BiLSTM-CRF on top of them, most of the computation is happening in the (fixed) large Transformer. Most NAS methods I'm familiar with re-learn fundamental aspects of the architecture (e.g., the Evolved Transformer), while fixing most of the architecture and re-learning a last layer or two is more suggestive of system combination or model stacking.

My main question is: did the authors try comparing to an ensemble or post-hoc combination of the predictions according to different models?  Computationally this would be cheaper than what the authors did. It's also much faster to search over 2^L-1 possibilities when checking each possibility just requires decoding the dev set rather than training the BiLSTM-CRF -- actually, this can be done very efficiently if each model's logits are cached.

There are more sophisticated variants of this like in the papers I linked above where each model has its own weights or additional inputs are used. Intellectually, I think these approaches are related, and they should be discussed and compared to.

RESULTS

As for the results, Table 1's gains are small -- they are consistent over random search, but I don't find them all that convincing.  There are too many embeddings here for ALL to work well -- my guess would be that a smaller set would yield better performance.

Tables 2-4 show improvements over existing baselines, XLM-R, and XLNet. This performance is commendable. However, again, I don't know how this compares to ensembling across a few leading approaches (like mBERT and XLM-R for the cross-lingual tasks).

CONCLUSION

In the end, I'm not sure how readily this approach will be picked up by others. Because the embeddings aren't themselves fine-tuned as part of the ensemble, it really feels more like a fine-tuned ensemble of existing models rather than true NAS. And the overhead of this approach is significant: it requires running many training runs over large collections of existing pre-trained models to get a small improvement over the current state-of-the-art. This is a possibly useful datapoint to have in the literature, but it feels like the technique isn't quite right to lead to more work in this area.

MINOR:

"use BiLSTM-Biaffine model (Dozat & Manning, 2017) for graph-structured outputs"

This is a very particular structure, namely a directed minimum spanning tree (MST), though projective trees are also possible using the Eisner algorithm. The paper should specify that it's these, and not arbitrary graphs that are being produced here.

-------------------

UPDATE AFTER RESPONSE

Thanks for the response and the additional experiments. The comparison between ACE and these other techniques is nice to see, although I'll note that both SWAF and voting shouldn't make totally independent predictions in tasks like NER, but should at least respect constraints in the label space (not sure if there were applied or not).

In the end, my opinion of this paper largely comes down to the practicality of this technique and its likelihood to be adopted more generally. This results in a large, complex model, and while I am now convinced that the authors have a better ensembling/combination technique than some others, I think it still falls short of a real "neural architecture search" contribution or a really exciting result.

---

> ### Author Response · Authors · 2020-11-22
> **Response to Reviewer #2**
>
> Thank you so much for your feedbacks. We really appreciate you taking the time out to share your thoughtful comments.
>
> ---
> ## For MOTIVATION/COMPARISONS:
> In model ensemble or model stacking, the individual models can predict the task outputs by themselves. However, the embeddings themselves are not independent models that can predict the task outputs and ACE targets at finding a single model to predict the outputs of the task. As a result, we believe ACE is quite different from the model ensemble or model stacking.
>
> We take your suggestion of comparing ACE with model stacking. We first tried SWAF [1] and voting in sequence labeling tasks and found that voting is stronger than training a meta-classifier for model stacking. Moreover, one of the benefits of voting is that it combines the predictions of the task models efficiently without any training process. We can search all possible $2^L-1$ model ensembles in a short period of time through caching the outputs of the models. However, searching the whole search space with SWAF is impractical. Training a single SWAF model requires about 5 minutes in our implementation for NER and searching for all possible ensemble requires about 7 days. We search for the best ensemble of models on the development set and then evaluate the best ensemble on the test set. We additionally search for the best ensemble on the test set for reference, which is the upper bound of the approach. Empirical results show that ACE is stronger than voting in all cases including searching over all possibilities of ensemble model on the test set. The All baseline is competitive with Ensemble model searching over all possible combinations on the development set. It is also stronger than the Ensemble model with all task models, which shows that the All baseline is strong enough. These results show the strength of embedding concatenation. Concatenating the embeddings incorporates information from all the embeddings and forms stronger word representations for the task model, while in model ensemble, it is difficult for the individual task models to affect each other through ensemble. Please refer to the first part of our general response to all reviewers and Section 5.3 in our revision for more details.
>
>
> ---
> ## For RESULTS:
> ### 1.
> As we discussed above, we have compared several approaches suggested by the reviewers. The empirical results show that the All baseline is competitive with all the other approaches. Although a smaller set for the All baseline might results in better accuracy, deciding the best smaller set requires high level domain knowledge of the task. What ACE does is automatically finding a smaller set that has better accuracy without such knowledge.
> ### 2.
> In our analysis, all possible $2^L-1$ combinations of ensemble models are searched including ensembling across the models with contextual embeddings. The results show that ACE outperforms ensemble approaches consistently.
>
> ---
> ## For CONCLUSION:
> Comparing with current state-of-the-art approaches, we believe the improvements cannot be considered as “small”. ACE outperforms current state-of-the-art approaches on a considerable scale in most of the tasks with a unified framework. Taking NER as an example, ACE is stronger than the current state-of-the-art proposed by [2] by 2.0 F1 scores and outperforms the All baseline by 1.1 F1 scores on average in our new analysis in Section 5.4 of the revision. For the speed concern, in practice the training speed is often less of a concern than the prediction speed, while the prediction of ACE is faster than that of the ALL baseline since some useless embeddings are filtered out after training.
>
> ---
> ## For MINOR:
> We use MST to ensure the tree output structure. We specified this in the Appendix A.3 of the revision.
>
> ---
> ## References:
>
> [1] Rajani and Mooney, Stacking With Auxiliary Features. In Proceedings of IJCAI 2017.
>
> [2] Yu et al. Named Entity Recognition as Dependency Parsing. In Proceedings of ACL 2020.

---

### Official Review · AnonReviewer4 · 2020-10-28
**AUTOMATED CONCATENATION OF EMBEDDINGS FOR STRUCTURED PREDICTION**

**Rating:** 6
**Confidence:** 4

**Review:**

Summary:

This paper proposes to automate the concatenation of word embeddings (obtained using different strategies) to produce powerful word representations for a given downstream task. To this end, the paper develops an approach based on Neural Architecture Search, wherein the search space is comprised of embedding candidates obtained using different concatenations. Using an accuracy-based reward function, it is showed that ACE can determine more effective concatenations. ACE is evaluated using  extensive experiments with different tasks and datasets, and it outperforms the two baselines (to different degrees) -- random search and concatenating all embeddings with no subselection.

##########################################################################

Positives:
- The idea of using NAS to construct concatenated embeddings is interesting and the formulation is clearly developed in the paper.
- The proposed approach is generic and can support different types of structured outputs (sequences, graphs etc.)
- The search process is computationally efficient and can be even run on a single GPU.
- A simple modification (based on a discount factor) is proposed to the reward function design that leads to non-trivial performance improvements.
- Strong experiment design: The proposed approach is evaluated on a large suite of datasets and tasks, and in many cases

Concerns:
- While the overall idea is interesting, the design choices made in the paper are not fully justified. Since ACE already pretrains the task model for each of the embeddings independently to begin with, why not adopt a "boosting" style approach instead of the naive "ALL" baseline. It is not surprising that even random search (known to be a strong baseline) consistently outperforms "ALL". The key challenge in concatenating disparate emebddings is that they can predict with varying degrees of confidence in different parts of the data and sequential inclusion of embeddings could be effective. In my opinion, the baselines chosen for concatenation are weak.
- Why is "accuracy" the best choice for reward design? There could be two different embeddings that could produce the same accuracy with varying levels of confidence (or empirical calibration). Unlike conventional ensemble learners, each contextualized representation is not a weak learner and hence it will be critical to take into account confidence estimates.

Overall, though the paper is experimentally strong, the design choices and the baselines need to be better justified.
##########################################################################

Questions during rebuttal period:

Please respond to the questions under concerns.

##########################################################################

---

> ### Author Response · Authors · 2020-11-22
> **Response to Reviewer #4**
>
> We want to thank you for your feedback and criticisms.
>
> ---
> ## For the concerns:
> ### 1.
> ACE does not pretrain the task model for each of the embeddings independently in the beginning. ACE trains a single model with different embedding concatenations at each step. At the first step, the task model of ACE is trained with all embeddings concatenated. We think that the description in Section 3.4 might be unclear. We have made it clearer in the revision.
>
> For the issue of ensemble models, we add an analysis in section 5.3 comparing the All baseline and ACE with ensemble models. For the ensemble learning, we use voting with confidences of models instead of other techniques due to the following reasons:
>
> (1) One of the benefits of voting is that it combines the predictions of the task models efficiently without any training process. We can search all possible $2^L-1$ model ensembles in a short period of time through caching the outputs of the models.
>
> (2) we tried SWAF [1] suggested by reviewer #2, and found it underperforms voting. Therefore, we believe that voting is a strong baseline of ensemble.
>
> In the first part of our general response, results show that ACE outperforms all the settings of ensemble model and even the ensemble approach searching over the test set. Moreover, the ensemble model with all task models is weaker than the All baseline in most of the cases, which shows that the All baseline is strong. These results show the strength of embedding concatenation. Concatenating the embeddings incorporates information from all the embeddings and forms stronger word representations for the task model, while in model ensemble, it is difficult for the individual task models to affect each other through ensemble. For more details, please check Section 5.3 in the revision and the first part of our general response to all the reviewers.
>
> ### 2.
> Since the target of ACE is finding a better concatenation of embeddings to achieve higher accuracy, we believe that using accuracy in the reward function design is a natural choice. Though it might not be the best choice, there is a lot of previous work (for example: [2], [3], [4]) that used accuracy in the reward function design to improve the model performance and got good empirical results. Taking the confidence estimation into account in the reward function design is a very interesting idea. We will try it in the future.
>
>
>
> ---
> ## References:
>
> [1] Rajani and Mooney, Stacking With Auxiliary Features. In Proceedings of IJCAI 2017.
>
> [2] Zoph and Le. Neural Architecture Search with Reinforcement Learning. In Proceedings of ICLR 2017.
>
> [3] Fan, et al. Learning to Teach. In Proceedings of ICLR 2018.
>
> [4] Qin et al. Robust Distant Supervision Relation Extraction via Deep Reinforcement Learning. In Proceedings of ACL 2018.

---

### Official Review · AnonReviewer3 · 2020-10-29
**reasonably interesting, though some concerns remain**

**Rating:** 6
**Confidence:** 4

**Review:**

Updates after discussion/revision period:

I think the revisions have improved the paper, but I'm not willing to increase my score or to fight for the paper. Overall, I think the paper represents a minor contribution, with its rigorous experimentation and some of its ideas, and that others may benefit from reading it, but I don't know that it is at the level of a typical ICLR publication.


--------

This paper describes an approach to choosing a subset of several options for (optionally contextualized) word embeddings to use in NLP tasks. Ideas are drawn from neural architecture search (NAS) and RL. The basic idea is to maximize dev set accuracy by searching over the space of embedding sets to use. There are some tweaks, including avoiding retraining-from-scratch for each set by keeping a single generalized model with all embeddings in it (where subsets can be chosen by setting some matrices to zero). Experiments are done on many NLP tasks (tagging, chunking, NER, parsing, etc.), leading to state-of-the-art results on nearly all test sets.

This paper represents an impressive number of experiments, considering several types of embeddings and many NLP tasks/datasets. It is well-written on the whole, though there are a few things I had confusion or concern about (details below). I lean positive on this paper, as it has an interesting algorithm that is more practical than prior work in NAS and has some promising results. However, I also have a few high-level concerns, described below:

1. I'm not sure if I'm thoroughly convinced of the empirical superiority of ACE. The primary baselines are All (using all embeddings always) and Random (random search over subsets of embeddings). Random is a little better than All on average, and ACE is a little better than Random on average. The average difference of 0.5 in Table 1 between ACE and Random is largely due to the 8 aspect extraction (AE) datasets, for which the differences are sometimes sizable. However, across the 17 other results in the table (tagging, NER, chunking, and parsing), none differ by more than 0.4, and the average difference between Random and ACE on those other 17 numbers is 0.17 (computed by me). Possibly statistically significant, especially because there are so many different datasets, but less impressive. If one were to deploy a method like this in practice, one would likely start by trying random search because it's so simple and doesn't require the slightly specialized learning framework and reward function in ACE.

Relatedly, I am concerned that the All baseline is not strong enough. Another natural baseline would be to start with All but then add a_l parameters as gates on the input embeddings, just as they are present in ACE. (The a_l parameters could be normalized to be between 0 and 1 by passing them each through a sigmoid before being multiplied with embedding vectors.) By having a single parameter to weight each embedding type in this way, the new version of All could switch on or off entire embedding types without adding many more parameters, which would make it more similar to the other methods. This would let us see the results of this stronger version of All (which, I would argue, is more likely to be used in practice than the current version of All).

2. My second concern is about the following sentence in Sec. 4.2: "If the pretrained contextualized embeddings are not available for a particular language, we use the pretrained contextualized embeddings for English instead." I find this to be a rather surprising decision, as it could add a great deal of noise for non-English languages. This could be especially problematic for the All baseline which doesn't have an easy way to switch off a noise type of embedding. It would be nice to know for which tasks/datasets this English embedding replacement was done in practice. I looked at Appendix A.4, but I wasn't able to determine from that section which embedding types were missing for which datasets.

3. CoNLL 2003 does not contain gold chunk labels. It contains automatic chunk labels (as can be confirmed by checking the original paper). CoNLL 2003 should not be used for chunking experiments. Unfortunately this mistake has been repeated in many papers. Please remove all CoNLL 2003 chunking experiments.

Some additional (less major) questions are below:

In Appendix B.2, why does ACE work better than retraining? I wouldn't have expected this to happen.

In Sec. 3.1, it's odd that the BiLSTM-CRF and BiLSTM-Biaffine functions only take in V as the only argument, where V is a function solely of the input x, not of the output y. Why is it not a function of y as well?

In Sec. 3.2, I find the phrasing "concatenation with the mask" to be a bit confusing. I don't think the mask is being concatenated; I think it's being multiplied elementwise with the embeddings.

Right above Eq. (7), there is the text "Taking m = 1" -- what is m?

In Sec. 4.2: What exactly is meant by "character embeddings"? There are many ways to embed words using characters. How are the character embeddings composed to form a word embedding?

How was the random search done? I see the sentence "For Random, we use the same training settings as our approach", which makes me assume there were 30 steps, but I think this should be made more explicit. Since the random approach and ACE are different algorithms with different hyperparameters, it's not clear to me what is meant by using "the same training settings".

---

> ### Author Response · Authors · 2020-11-22
> **Response to Reviewer #3**
>
> Thank you for the great review! We really appreciate you taking the time out to share your thoughtful comments.
>
> ---
> ## For the concerns:
>
> ### 1.
> Though the design of ACE is slightly specialized, we believe that the algorithm framework is general over different tasks and the implementation is not such difficult. If anyone wants to achieve higher accuracy on a specific dataset, ACE is fast and effective comparing with Random.
>
> For the All baseline, we follow your suggestion to weight each embedding in the All baseline. We conduct the experiment on one of the datasets for each task for efficiency, which is the same setting as Appendix B.2. Results (in Section 5.3 of the revised paper) show that All+Weight is stronger than All baseline by 0.06 on average. However, All+Weight is not consistently stronger than All baseline in all of the tasks. One possible reason is that the linear layer in Eq. 2 is able to implicitly represent the weight of each embedding. Therefore, All+Weight is possibly a stronger version of All baseline but the improvement is limited. Please check Section 5.3 in our revision and the first part of our general response to all reviewers for more details.
>
> ### 2.
> We use this setting for the following reasons.
>
> (1) we want to make the search spaces of most of the datasets having almost the same size so that the hyperparameter settings (especially the running iterations) are unchanged.
>
> (2) we want to see whether the embeddings pretrained with a large corpus of other languages (i.e. the English embeddings) are able to transfer to another language to improve the accuracy.
>
> More specifically, we applied this setting on the Turkish and Russian datasets in aspect extraction since we didn’t have Flair and ELMo embeddings for Turkish (“tr“) and we didn’t have language-specific BERT, Flair and ELMo embeddings for Russian (“ru“). We run the All baseline for “tr“ and “ru“ without English embeddings and get 69.7 and 70.2 for “tr“ and “ru“ respectively. The results are higher than the All baseline in Table 1 but are still inferior to ACE. In the searched embedding concatenation of ACE, English BERT is selected in all runs for “ru“ while English ELMo is not selected in either “tr“ or “ru“. The English Flair embeddings is selected in a majority of the runs. The observation shows that the pretrained sub-token and character information in English BERT and Flair might be helpful to the tasks of other languages with limited embedding resources. On the other hand, the pretrained word information in English ELMo might add noises to the word representations on these tasks. We have made the setting clearer in the revision.
>
> ### 3.
> According to your suggestion, we removed all the experiments of CoNLL 2003 chunking experiments in the revision. We split 10% of the training set of CoNLL 2000 as the development set instead and rerun the experiments. Please check our revision for more details.
>
> ---
> ## For the additional questions:
>
> For 1, one possible reason is that the retrained models are randomly initialized while ACE models are initialized by the parameters of trained model of previous step.
>
> For 2 – 4, we have revised the paper according to your comments.
>
> For 5, "character embeddings" is the non-contextual character embeddings proposed by [1].
>
> For 6, you are right. The random search is done by searching 30 steps. We have made it more explicit in the revision.
>
>
>
> ---
> ## References:
> [1] Lample et al., Neural Architectures for Named Entity Recognition. In Proceedings of NAACL 2016.

---

### Author Response · Authors · 2020-11-22
**General Response to all reviewers:**

We thank all the reviewers for their time and valuable feedback. We revised the paper with the feedback and added two new analyses in the paper.

---
## 1.
The first analysis answers the reviewers’ the main concern that the All baseline might not be strong enough. In Section 5.3, we conduct a comparison among ACE, All and two additional approaches suggested by the reviewers.

The first additional approach is a variant of the All baseline, which is weighting each embedding with a learnable parameter (suggested by Reviewer #3).

The second additional approach is ensemble of the task models, which is mentioned by Reviewer #1, #2 and #4. Specifically, we train one model with each embedding candidate and then combine the predictions from all the trained models through an ensemble algorithm. We tried to implement the approach of “Stacking With Auxiliary Features” (SWAF) [1] suggested by Reviewer #2, which is a model stacking approach. We trained a meta-classifier to predict the outputs based on the prediction and confidence of task models and word features as input. We first tried the ensemble model for sequence labeling tasks (NER, POS, Chunking, AE). The model predicts the label at each position separately. The meta-classifier is trained on the development set and tested on the test set. However, we found that the accuracy improvement of the ensemble model is moderate comparing to the best task model. Besides stacking, we also tried voting with confidence, which is much simpler. We found that its accuracy is higher than stacking. As a result, we use voting as the ensemble approach in the new experiments in the revision. Here is a comparison between voting and SWAF in sequence labeling with an ensemble of all task models:

|Approach| NER | POS | AE | Chunk |
|-|-|-|-|-|
|SWAF | 91.8 | 89.0 | **68.1**| **96.5** |
|Voting | **92.2** | **90.6** | **68.1** | **96.5** |

One of the benefits of voting is that it combines the predictions of the task models efficiently without any training process. We search all possible $2^L-1$ model ensembles in a short period of time through caching the outputs of the models (Searching the whole search space with SWAF is impractical. Training a single SWAF model requires about 5 minutes in our implementation for NER. Therefore, searching for all possible ensemble requires about 5$\times$2047 minutes $\approx$ 7 days). Therefore, we search for the best ensemble of models on the development set and then evaluate the best ensemble on the test set. Moreover, we additionally search for the best ensemble on the test set for reference, which is the upper bound of the approach.

We conduct the experiment on one of the datasets for each task for efficiency, which is the same setting as Appendix B.2. Empirical results show that ACE outperforms all the settings of these two additional approaches and even the ensemble approach searching over the test set, which shows the effectiveness of ACE. The All baseline is competitive with the two additional approaches in most of the cases.
There is no clear winner of these approaches on all the datasets, which shows that All is as strong as the two additional approaches in our experiments. Additionally, the All baseline is stronger than the ensemble model with all individual task models. These results show the strength of embedding concatenation. Concatenating the embeddings incorporates information from all the embeddings and forms stronger word representations for the task model, while in model ensemble, it is difficult for the individual task models to affect each other through ensemble.



---

## 2.
The second analysis is about our new results on NER. Recently, models with document-level word representations extracted from transformer-based embeddings significantly outperform models with sentence-level word representations on NER tasks [2], [3]. To show the effectiveness of ACE with document-level representations, we replace the sentence-level word representations from transformer-based embeddings (i.e., XLM-R and BERT embeddings) with the document-level word representations.

Empirical results show that the document-level representations can significantly improve the accuracy of ACE. Comparing with models with sentence-level representations, the averaged accuracy gap between ACE and the All baseline is enhanced from 0.7 to 1.1 with document-level representations, which shows that the advantage of ACE becomes stronger with document-level representations. Please refer to Section 5.4 for more details.

---
We hope the two analyses will help the reviewers to better understand the strength of ACE.

---
# References:

[1] Rajani and Mooney, Stacking With Auxiliary Features. In Proceedings of IJCAI 2017.

[2] Devlin et al. BERT: Pre-training of Deep Bidirectional Transformers for Language Understanding. In Proceedings of NAACL-HLT. 2019.

[3] Yu et al. Named Entity Recognition as Dependency Parsing. In Proceedings of ACL 2020.

---

### Author Response · Authors · 2020-11-22
**Detailed Results:**

Due to the character limitation of comment, here we post the results of our two additional analyses for your information:

---
## Analysis 1 (Section 5.3)

|Approach|NER|POS| AE| Chunk|DP-UAS|DP-LAS|SDP-ID|SDP-OOD|
|-|-|-|-|-|-|-|-|-|
|All|92.4 |90.6 |73.2 |96.7 |96.7 |95.1 |94.3 |90.8|
|Random|92.6|91.3|74.7|96.7| 96.8|95.2 |94.4 |90.8 |
|ACE|**93.0**| **91.7**| **75.6**|**96.8**| **96.9**| **95.3** |**94.5** |**90.9**|
|All+Weight|92.7 |90.4 |73.7 |96.7 |96.7 |95.1 |94.3 |90.7|
|Ensemble |92.2 |90.6 |68.1 |96.5 |96.1 |94.3 |94.1 |90.3|
|Ensemble$_\text{dev}$|92.2 |90.8 |70.2 |96.7 |96.8 |95.2 |94.3 |90.7|
|Ensemble$_\text{test}$|92.7 |91.4 |73.9 |96.7 |96.8 |95.2 |94.4 |90.8|

---
## Analysis 2 (Section 5.4):
+sent/+doc: models with sentence-/document-level embeddings

|Approach | de | de$_\text{06}$ | en | es | nl |
|-|-|-|-|-|-|
|All+sent | 86.8 | 90.1 | 93.3 | 90.0 | 94.4 |
|ACE+sent | 87.1 | 90.5 | 93.6 | 92.4 | 94.6|
|BERT [1] | - | - | 92.8 | - | - |
|Akbik et al. (2019) [2] | - | 88.3 | 93.2 | - | 90.4 |
|Yu et al. (2020) [3] | 86.4 | 90.3 | 93.5 | 90.3 | 93.7 |
|All+doc | 87.6 | 91.0 | 93.5 | 93.3 | 93.7 |
|ACE+doc | **88.0** | **91.4** | **94.1** | **95.6** | **95.5** |

---
## References:

[1] Devlin et al. BERT: Pre-training of Deep Bidirectional Transformers for Language Understanding. In Proceedings of NAACL-HLT. 2019.

[2] Akbik et al. Pooled Contextualized Embeddings for Named Entity Recognition. In Proceedings of NAACL-HLT. 2019.

[3] Yu et al. Named Entity Recognition as Dependency Parsing. In Proceedings of ACL 2020.

---

> ### Comment · Area_Chair1 · 2020-11-23
> **Quick questions**
>
> Dear authors,
> Thanks for the response. Would be great if you could quickly address the following points:
> 1. A simple baseline is to choose a single embedding based on a validation set and use just this one. This would correspond to an ensemble of one in Table 6. Since there is a gap between Ensemble_dev and Ensemble_test, it seems like there is some overfitting
> in the choice of ensemble and therefore reducing the search space could improve things (assuming there is a single good embedding). Have you tried this?
> 2. For table 6, did you use the same validation set for ACE and the other methods?
> 3. The concatenated models seem to be using more parameters than the baseline models (i.e., using a single embedding). Thus, it seems to make sense to compare to a single embedding model that has the same number of parameters as the one with concatenated embeddings. Have you explored this?
> Thanks

---

> > ### Author Response · Authors · 2020-11-24
> > **Response to the questions**
> >
> > Thanks for your great questions!
> >
> > ---
> > ### 1.
> > Here is the test accuracy of single embedding models with the highest development set accuracy for each task.
> >
> > |Approach|NER|POS| AE| Chunk|DP-UAS|DP-LAS|SDP-ID|SDP-OOD|
> > |-|-|-|-|-|-|-|-|-|
> > |Best Single Model|92.0|89.7|73.7|96.5|96.8|95.2|94.1|90.3|
> >
> > By comparing these results with the results in Table 6 (or the first table above), we can see that the accuracy of single embedding models is not stronger than that of Ensemble$_\text{dev}$ in most of the cases except AE. But the accuracy of AE is still inferior to Ensemble$_\text{test}$ and ACE.
> >
> > ---
> > ### 2.
> >
> > Yes, we use the same dataset settings in all the experiments of Table 6. Therefore, the validation set of ACE and the other approaches is identical.
> >
> > ---
> > ### 3.
> > There are two conditions that can be considered as “the same number of parameters”:
> >
> > (1)	The number of parameters of the model including all the embeddings is the same.
> >
> > (2)	The number of trainable parameters of the model is the same. More specifically, the number of parameters excluding non-trainable embeddings.
> >
> > For (1), it is impractical to train such a model as the All baseline has more than 1 billion parameters in total. It possibly requires dozens of GPU to train such a model efficiently.
> >
> > For (2), following your suggestions, we build a deeper trainable network by adding a linear layer to project the embeddings to the same hidden size as the concatenation of all embeddings for the best single embedding model (which is ELMo) in NER. The number of trainable parameters of the resulting task model is larger than that of the model with all embeddings concatenated since there is an additional $k \times d$ weight in the model, where $k$ is the hidden size of the single embedding and $d$ is the hidden size of concatenated embeddings. With the linear projection layer, the accuracy of NER reduces from $92.0$ to $91.9$. The accuracy is still inferior to the accuracy of the All baseline and ACE, which is $92.4$ and $93.0$ on average in CoNLL NER respectively. For the reduction of accuracy, one possible reason is that additional parameters make the task model difficult to train. We also tried some smaller hidden sizes $d$ for the projection, but we did not find further improvement.
> >
> > In conclusion, the cause of the accuracy gap between a single embedding model and the concatenated embeddings model is possibly not due to the number of model parameters.

---

### Decision · Program_Chairs · 2021-01-07
**Final Decision**

**Decision:**

Reject

**Comment:**

The paper proposes a method for using multiple word embeddings in structured prediction tasks. The reviewers shared the concerns that the method seems rather specific to this use case and the empirical improvements do not justify the complexity of the approach. They also questioned the definition of the method as "architecture search" vs a particular ensembling method.
Finally, I think the authors should provide more discussion of why using all the embeddings (in the sense of bias-variance tradeoffs).